# Poly(Vinyl Alcohol) Recent Contributions to Engineering and Medicine

**Dorel Feldman**

Faculty of Engineering and Computer Science, Concordia University, 1455 De Maisonneuve Blvd.West, EV-6-403, Montreal, QC H3G 1M8, Canada; dorel.feldman@concordia.ca; Tel.: +1-(514)-848-2424

**Abstract:** Poly(vinyl alcohol) (PVA) is a thermoplastic synthetic polymer, which, unlike many synthetic polymers, is not obtained by polymerization, but by hydrolysis of poly(vinyl acetate) (PVAc). Due to the presence of hydroxylic groups, hydrophilic polymers such as PVA and its composites made mainly with biopolymers are used for producing hydrogels that possess interesting morphological and physico-mechanical features. PVA hydrogels and other PVA composites are studied in light of their numerous application for electrical film membranes for chemical separation, element and dye removal, adsorption of metal ions, fuel cells, and packaging. Aside from applications in the engineering field, PVA, like other synthetic polymers, has applications in medicine and biological areas and has become one of the principal objectives of the researchers in the polymer domain. The review presents a few recent applications of PVA composites and contributions related to tissue engineering (repair and regeneration), drug carriers, and wound healing.

**Keywords:** poly(vinyl alcohol); hydrogels; composites; applications in engineering and medicine

## 1. Introduction

Poly(vinyl alcohol) (PVA) is a thermoplastic polymer that is obtained by the hydrolysis of poly(vinyl acetate) (PVAc) and not by polymerization processes like some other synthetic polymers. After hydrolysis, PVA still contains 1–2 mol% of acetyl groups. Its degree of polymerization (DP) depends primarily on the size of the PVAc macromolecular chain. The transformation of PVAc into PVA is obtained by the base catalyzed alcoholysis or by the acid initiated hydrolysis. PVA is the most polar synthetic polymer, it is odorless, nontoxic, biocompatible, and soluble in water, acids, and high polar solvents. Its molecular weight (MW) depends on PVAc MW and the degree of hydrolysis.

PVA is well known in its application in the production of fibers including its use in surgeries, artificial leather, tubing, gaskets with good stability to oil derivatives, rubber-like items, transportation belts, emulsifiers, adhesives for paper and paperboard, and in general purpose adhesives for bonding paper, textiles, leather, and porous ceramic surfaces. When processed for textile fibers and other applications, it should be made water insoluble. For PVA fiber, this is done with an aqueous solution of sodium sulfate containing sulfuric acid and formaldehyde. This treatment transforms the hydroxyl groups into cyclic formal groups.

Being a water soluble synthetic polymer, if no measures are taken against high temperature, light exposure, microorganisms, etc., PVA may degrade in the environment by photodegradation, bio degradation, and chemical degradation; the latter includes hydrolytic and oxidative processes.

A group of researchers studied the rheological behavior of the aqueous solution of PVA with different MW and concentration subjected to freeze–thaw. The experience results suggest that the number of PVA segments participating in the crystalline junction points increase exponentially during the freezing, while decreases exponentially during thawing in the vicinity of the critical point [1].

This review presents the recent studies conducted for the synthesis and application of different PVA products and PVA/bio-polymer composites in various engineering and medicine fields.

### 1.1. Poly(Vinyl Alcohol) (PVA) Hydrogels

The Table 1 contains a few PVA composites and their applications that will be discussed in the review.

**Table 1.** Examples of composites and their applications.

| PVA Hydrogel | Bio-or Synthetic Polymer as Second Component | Application | Reference |
|---|---|---|---|
| PVA/Agar | Agar | As Biocompatible, and Bioactive | [2] |
| PVA/CS/PDA-GO | Chitosan | Adsorption Ion elements | [3] |
| PVA/alginate | Alginate | Fertilizer | [4] |
| PVA/PDMAEMA-PAA | PDMAEMA | Sensors, tissue eng | [5] |
| PU/PVA | PU | Drug carrier | [6] |
| PVA/Pu/L/Gl | Lysine, Gelatin | Wound dressing | [7] |
| PVA/chondroitin sulfate | Chondroitin | Bone tissue | [8] |
| PAA/Fe$^3$/Gl/PVA | Gelatin | Biomedical | [9] |

Hydrogels made of synthetic polymers such as PVA have gained much attention in the last few decades due to their physico-mechanical features. Hydrogels are three-dimensional (3D) networks of cross-linked hydrophilic macromolecules with high water content (up to 90%) and are highly elastic and soft. Hydrogels are characterized by tunable physical, chemical and biological properties, biocompatibility, low toxicity, and good swelling behavior, which make them promising materials with applications in different fields [10]. A great advantage in using hydrogels is the possibility of adapting their characteristics to match a specific application.

Poly(acrylic acid)-Fe$^{3+}$/gelatin/PVA (PAA-Fe$^{3+}$/Gl/PVA) triple network supramolecular hydrogels were synthesized and exposed to cooling and freezing/thawing. Healed PAA-Fe$^{3+}$(0.20)/Gl 3%/PVA 15% triple network hydrogels sustain a tensile strength of 96.6% of the tensile strength of the original sample [11].

After soaking in ammonium sulfate solution, the PVA/agar hydrogel bio-composite becomes dense and uniform with stronger H bonds between the polymers; the tensile strength and toughness increased to 18.0 MPa and 42.3 MJ/m$^3$, respectively. The polymers and the easy-operating technique provide promising applications as biocompatible and bio-active materials [2].

Due to PVA biocompatibility, composites made of PVA and bio-polymers such as cellulose, chitosan, gelatin, casein, and others with different characteristics have found new interesting applications. PVA bio-composites with regenerated cellulose softwood pulp (RC-SP) as a green reinforcement were prepared via co-precipitation. Simultaneous co-precipitation promotes uniform dispersion of the RC-SP and constructs strong molecular chain entanglements and H bonding network inside the composites. The physical cross linking network reduces the water absorption and improves the water resistance. Due to the strong filler–matrix interaction, the composite has a higher thermal decomposition characteristic and better mechanical and dynamic mechanical properties [12].

Polyblends PVA/casein (CA) were prepared using the solution-casting technique. The tests done on this composite showed interaction between the two polymers and that the tensile strength of CA increased with the amount of PVA. Such films have the potential to be used in biodegradable packaging applications [3].

Polydopamine-functionalized graphene oxide (PDA-GO) was used to form PVA/chitosan/PDA-GO (PVA/CS/PDA-GO) hydrogels. The adsorption of metal ions like Cu(II), Pb(II), and Cd(II) onto the new

PVA/CS/PDA-GO hydrogel beads with pH variation has been studied. These hydrogel beads could be a potential recyclable adsorbent for the removal of some hazardous metal ions in waste water [13]. Another study evaluated the performance of PVA/bentonite hydrogel and its freezing/thawing as an efficient dye remover [14].

By incorporating urea in a PVA-alginate hydrogel core followed by $HCO_3^-/CO_3^{2-}$ rich alkaline cell-free ureolytic culture broth, a slow release fertilizer was obtained [15].

The properties of the hydrogels already mentioned make them very useful in many applications in engineering fields such as packaging, the removal of hazardous metal ions from wastewater, gas separation membranes, and in the electrical and electronic domain. As will be presented later on, hydrogels are also used in medicine for wound dressing, antibacterials, tissue engineering, as drug carriers, and implants, etc. The hydrogels made of composite PVA/bio-polymers and their nanocomposites that have already begun to be studied and, at a small scale, will increase their possibilities for use in various domains.

### 1.2. Other PVA Composites

In the case of PVA/sodium alginate (SA), the total dipole moment (TDM), HOMO/LUMO energy gap, and electrostatic potential were calculated. The results indicated that the TDM increased, the HOMO/LUMO energy gap decreased, and electro-negativity increased.

As the result of the interaction between PVA and sodium alginate (SA), the total dipole moment increased. Thermal parameters showed a variation of changing the site of interaction, which indicated that the coordination of PVA/SA is an important factor for describing this kind of composite [4].

The incorporation of pomegranate peel powder (PPP) significantly affected the surface, morphology, physical, mechanical, and barrier properties of the PVA based films. A higher PPP amount leads to the production of a flexible and stretchy film and better light barrier properties.

PVA/PPP films have great potential as a green packaging product for cosmetics [16].

Supercapacitors have attracted much interest due to their high power density and long cycling life. However polypropylene membranes that are widely used as separators in supercapacitors are unfavorable for transporting electrolyte and electrodes due to their hydrophobic characteristics. As a consequence, a cross-linked solid polymer electrolyte membrane and a semi-interpenetrating polymer network were fabricated from sulfonated poly(ether ether ketone) and PVA which can be used as hydrophilic separators [17].

Polymer based composites are extensively used in the electronic industry for energy storage applications like conventional capacitors, supercapacitors, batteries, and fuel cells due to their superior mechanical and electrical properties. In the case of conventional capacitors, the polymer composites are generally employed as a dielectric layer. Oxide semiconductors incorporated polymer composites are widely used as dielectrics for flexible capacitors [18] PVA is an dielectric; its compatibility with other polymers like poly(3,4-ethylenedioxythiophene):poly(styrene sulfonate) allows for its use in the electrical field. Being a polar polymer makes it useful for semiconductors.

Using solution casting, composite PVA films with various amounts of $Cu(NO_3)_2.3H_2O$ were prepared. It was found that the alternating current (AC) conductivity increased by increasing the dopant concentration up to 10 wt% [19].

The direct current (DC) conductivity of PVA hydrogels and ferrogels crosslinked with borax has also been studied. The increase in the amount of borax led to the rise in the DC conductivity value, but the increase in the carbonyl iron dopant with an unchanged borax amount led to a decrease in the conductivity [20].

Using PVA, poly(vinyl pyrrolidone) (PVP) and lithium acetate, lithium ion conducting polymer blend electrolytes were prepared. A higher electrical conductivity was determined for the polymer blend with 50 PVA; 50 PVP; 25% lithium acetate at 303 and 363° K. The dielectric and loss tangent analysis were also carried out for the prepared polymer electrolyte and the higher conductivity sample

at different temperatures. The transference numbers of polymer electrolytes was calculated by Wagner's polarizing technique and confirmed by the Bruce–Vincent technique [21].

Surface modification of fillers is used to reduce interfacial thermal resistance in thermal conductive products. Graphite (G) was modified in a research with acrylic acid (AA) to decrease its inertness (AA@G). The thermal conductivity of AA@G exhibited extremely high enhanced efficiency. A series of PVA composites, some loaded with AA@G and others with G were produced in order to study the interfacial interaction of the matrix–filler. With the synergistic increment in thermal conductive performance, AA@G exhibited a high enhanced efficiency of 358% compared with the literature data [22].

Incorporation of fillers into gel electrolytes has been used to improve the electrochemical performance of energy storage devices. Graphite oxide (GO) gel electrolytes were prepared with PVA and $H_3PO_4$ as an ion producer. The GO containing gel has superior mechanical properties, which makes it a potential candidate for use as a gel electrolyte in flexible and wearable electronic gadgets [23].

A PVA solution subjected to freezing–thawing was added to magnesium acrylate, leading to the formation of a Mg acrylate/PVA interpenetrating network, which contains, besides the acrylate, the PVA network based on H bonds; its fracture stress was 1.44 MPa and self-healing efficiency was 80% after 3 h [24].

A study showed that the addition of a plasticizer could modify the structure of PVA and after such addition, the crystallization energy increased. The study indicated that the plasticizers were able to modify the original crystalline structure of PVA and interacted with PVA to form stronger H bonds to replace the intermolecular phase of PVA intramolecular interactions [25].

Poly(vinyl butyral) (PVB) with different content of acetal groups is obtained in water by the condensation reaction between two OH groups of PVA and butyraldehyde in the presence of an acid catalyst.

PVA/water systems used in PVB production are prone to structuring with consecutive interaction with consecutive formation of strong interchain interactions. PVB is characterized by a unique set of adhesive and binding properties as well as the high strength of the fiber and film materials. Plasticized PVB is used as a laminating film in the production of safety glasses (triplex), and as an adhesive in the production of varnishes, primers, enamels, and mastics [26].

The effect of the addition of pentaerythritol to PVA on the swelling, mechanical, and optical properties were investigated. Elongation at break increased, tensile strength decreased, and the optical performance of the PVA films was improved [27].

A one pot two-step process in the case of microwave irradiation has been used to fabricate PVA sponges and PVA/phytic acid (PVA/PA) polymer sponges. Their characteristics such as thermal stability, surface resistivity, and self-extinguishing time led to the following data: 235.5 and 354.8 °C, $540 \times 10^9$ and $1.63 \times 10^5$ Ω/sq, and 18.00 and 9.00 s, respectively [28]. The sponges can be used for insulators, electrodes, scaffolds for medical applications, and replacements for infected bone tissue.

Three-dimensional porous PVA/CS sponges with good mechanical strength, high absorption capacity, and recycling ability is of scientific and technological interest and can have industrial applications such as dehydration of acetone/water, water purification, and as fuel cells and batteries [29].

An organic phosphonic acid (OPA) was selected as the protic media to produce phosphonated proton exchange membranes based on PVA. Proton conductivities and methanol permeability through the membrane were investigated in terms of various amounts of OPA [30].

4-Formyl dibenzo-18 crown-6 grafted PVA as anion exchange membranes were prepared. After alkali treatment, a conductivity degradation rate less than 8.6% reflected high alkali resistance. The test results showed that the membranes complied with the requirements in the anion exchange membrane fuel cell [31].

Composite anion exchange membranes based on imidazolium and quaternary ammonium-functionalized PVA were used for direct methanol alkaline fuel cell application. It showed a superior performance compared to the quaternary ammonium functionalized membrane [32].

A review covered recent studies on PVA-based proton exchange membrane in different fuel cell applications including proton-exchange membrane fuel cells and direct methanol fuel cells [33].

New iota-carrageenan–g-PVA polyelectrolyte membranes developed for application in direct methanol fuel cells have also been studied. The ion-exchange capacity, water, methanol sorption, and the methanol crossover flux across the polyelectrolite membranes were adopted as monitors for this research. The efficiency factor for the prepared membrane was one order higher than that of Nafion 117 [34].

The polyblend PVA/sodium polyacrylate was used to form a membrane for $CO_2$ separation [35].

Films have been made based on a PVA (with plasticizers and cross linkers) matrix and banana pseudostem fiber. At 20% fiber amount, the flexibility of the films was high with the elongation at break more than 100% and tensile strength of 30.8 Mpa, which is close to the commonly used Low-density polyethylene (LDPE) package films. With alkali treated banana pseudostem fibers, the films had a maximum tensile strength of 34.2 MPa and lower water uptake of only 60% [36].

One study focused on the effects of palm oil fuel ash (POFA) and PVA on the physico-mechanical, thermal, and morphological properties of a kenaf-jute reinforced PVA/PE hybrid bio-composite. TGA and DSC results confirmed that the jute/kenaf-PE/POFA composite had a higher thermal decomposition and activation energy and more stability than the jute/kenaf-PE and jute-kenaf-PE/PVA composites, which is recommended for reinforcing concrete [37].

Films based on PVA and keratoses were prepared in water. The analysis pointed out that the two polymers had no covalent interaction with each other. Increasing the amount of PVA in the polyblend film from 70 to100% improved its tensile strength and the elongation at break. It appears as a promising candidate for the producing of a new biocompatible material appropriate for different applications ranging from medical to filtration and adsorption equipment [38].

A research has been done on the synthesis and optimization of a green PVA-co-poly (methacrylic acid) adsorbent. It was found that it possess the maximum adsorption capacity of $0.761\,mg \cdot g^{-1}$ at the equilibrium of methylene blue dye from an aqueous solution at 10 ppm concentration; 500 mg of sample done at pH = 7 and 30 °C. The adsorbent exhibits regeneration efficiency for four successive adsorption-desorption cycles [39].

In a study, highly water selective PVA/polyacrylonitrile (PAN) pervaporation membranes were prepared. The pervaporation performances were investigated by separation of 95 wt% ethanol aqueous. After running for 120 h, the membranes still displayed a good stability for the ethanol dehydration in the pervaporation process [40].

Experiments of arsenic removal (i.e., As(III) and As(V) anions) indicated that PVA/ZnO with a maximum removal of 97% is a highly efficient sorbent for this element [41].

A new easy synthesis approach using PVA for producing highly uniform tough crosslinked poly(methyl methacrylate) micro particles for more potential applications has been developed for fabricating highly uniform crosslinked PMMA microparticles with the desired mechanical strength to meet the requirements of electrical packaging, for instance, applied as anisotropic conductive films or spacers for liquid crystal display assembly [42].

Compared with non-plasticized PVA fiber, the plasticized PVA with pseudo ionic liquids and glycerol provided a lower melting temperature, wider thermal processing interval, and better melt fluidity. The PVA plasticized fibers were successfully produced with the melt spinning techniques. The melt-spun PVA fibers would have potential applications as a fiber reinforcing product in the concrete field [43].

A new technique was used to obtain an isotactic polypropylene (PP)//PVA composite. This consisted of introducing PVA aqueous solution into molten PP. The PVA fibers oriented in the PP flow direction provided interesting properties like high modulus, high yield stress, and bending deformation as well as elevated heat distortion temperature [44].

The study of the PVA fiber–matrix interactions in composites showed that polar surface functionalities led to a strong adhesion while nonpolar hydrophobic surface layers decreased the adhesion [45].

In one study, the kinetic and structural characterization over time of PVA microgels obtained through a salting-out process were presented. The micro particle preparation based on a salting-out process constitutes a novelty if we compare it to the methods already existing and based on the use of crosslinkers and is a cheap and easy protocol that allows tuning both particle size and density by varying the salt concentration [46].

## 2. PVA Contributions to Medicine

The application of natural and synthetic polymers to medicine has become one of the principal objectives facing polymer researchers. Hydrogels of natural or synthetic polymers have many medical and biological applications; they have been used more in recent studies. Bioactive factor delivery from biopolymer hydrogels provides a versatile approach to treat diseases. Additionally, a hydrogel with good mechanical strength is desirable in applications such as sensors and tissue engineering, being one of the most attractive biomaterials for regenerative engineering. As drug delivery hydrogels have the advantage of being stimuli-responsive, they are considered as smart polymers. Many of them respond to chemical and physical stimuli such as ionic strength, pH, temperature, light, and electrical, and magnetic fields [47].

In another study, a PVA/poly(2-NN'-dimethylamino) ethyl methacrylate (PDMAEMA)–PAA hydrogel was produced. The double-network PVA/PDMAEMA)-PAA hydrogel exhibited satisfactory mechanical strength; a tensile strength of 0.45 MPa and compressive strength of 1.2 MPa was obtained for the double network hydrogel and the compressive strength was 480% higher than that of the single-network hydrogel. The study contributes to broadening the application of these kinds of hydrogels as sensors and tissue engineering [5].

The optimal amounts of the hydrogel components established in a recent study were 12% PVA, laccase concentration of 836 μg/mL, and ferulic acid concentration of 1.95 mM. The results of this study confirmed the PVA reticulation with ferulic acid and the presence of crosslinks among the PVA macromolecules [48].

The aid of hydrogels in biomedical application is limited by swelling and weak strength under physiological conditions. By using methacrylated PVA and thiol terminated PVA, a non-swelling hydrogel was obtained, which was non-toxic to L929 cells, which is favorable for promising biomedical applications as implants and tissue engineering scaffolds [49].

An overview of the current gelling techniques discussing in detail the state-of-the-art of various synthesis methods and biomedical applications of various hydrogels is presented. This shows that the field of tissue engineering places complex demands on biomaterials including polymers that are applied for organ/tissue development and repair. Future biocompatibility and cytotoxicity tests should be carried out to establish the potential application of polymer hydrogels for biomedical purposes [50].

By freezing–thawing, polyurethane/PVA (PU/PVA) hydrogels have been obtained. Their tensile strength was lower than that of the PVA hydrogel due to the extent of the occurrence of H bonding during freezing–thawing cycles, which determines the variation of porosity. Through the application of the in vitro technique, the hydrogel was used for release of the drug neomyein sulfate. The study showed that PU/PVA hydrogels can be used as drug carriers [6].

A lyophilized hydrogel composite containing alginate, gelatin, and PVA was produced with the aim of absorbing exudates, maintaining a moist environment, and enhancing interaction with the tissues. In the scaffold, triiodothyronine was introduced due to its vital role in the repair and regeneration of tissues. The researchers considered that the produced scaffold possessed a great potential as a chronic wound therapeutic [51].

With the aim of achieving enhanced wound healing, PVA/pullulan/poly L-lysine/gelatin (PVA/Pu/L/Gl) hydrogels were produced. The overall results of this study showed the potential of the PVA/Pu/L/Gl hydrogels for the application as a wound dressing [7].

A malleated PVA (PVAM) grafted copolymer with anionic polyacrylamide (PAM) was synthesized (PVAM-g-PAM) for use in drug encapsulation [52].

Amoxicillin-loaded films of PVA-g-PAM of varying composition were recently prepared. Aside from antibiotic applications, the drug loaded grafted hydrogels were also examined for effectiveness against Gram-negative bacteria [53].

PVA/chondroitin sulfate hydrogel scaffolds was prepared by using glutar aldehyde as a crosslinking agent for the regeneration of articular cartilage. PVA increased the bio-adhesiveness and mechanical properties. Chondroitin sulfate increased the content of glucosaminoglycan in the extra cellular matrix [8].

The aim of one study was the design of a 3D scaffold composed of PVA. The scaffold was produced using a combination of gas foaming and freeze–dry processes that did not need any crosslinking agent. The developed scaffold shows potential for use as a biomaterial for cardiac tissue engineering applications [54].

Osteoarthritis is a generative joint disease of the articular cartilage and extends to the subchondral bone. The interface between these soft and hard tissues has a significant role on osteoarthritis. In this research, PVA/Gl hydrogels were prepared by chemical and freeze–thaw structural formation physical crosslinkage. The PVA/Gl ratio of 79:30 demonstrated suitable structural formation, physical properties, and biological functions to induce tissue formation at the subchondral bone interface for osteoarthritis surgery [9].

PVA/CS composite hydrogels were prepared by the synergistic effect of H bonding, crystallization, chain entanglement, and ionic interactions without the addition of harmful chemicals. Due to a relatively homogeneous network and high crosslinking density, the gel exhibited very good mechanical properties, antiseptic, electrical conductivity, and swelling-resistant ability. The authors hope that these results will enable the development of tissue engineering materials for commercial applications [55].

Another study considered for the first time the use of halogens as less aggressive agents than potassium permanganate to perform PVA oxidation. Scaffolds were assessed for their mechanical properties and cell/tissue biocompatibility through the cytotoxic extract test. The halogens were successfully produced in the effort of adopting polymer characteristics to specific tissue engineering applications [56].

## 3. Remarks

This article reviewed new recent research contributions based on PVA and its composites in engineering and medicine. The results of the studies done recently (2019, 2020) in these domains and presented in this review could lead to new composites of PVA with synthetic or natural polymers with new interesting properties and applications. More comprehensive studies are required to increase the basic knowledge on hydrogels, which with their excellent characteristics like biocompatibility, swelling in different media, sensitivity to temperature, pH, ionic strength, light, electrical and magnetic fields, other stimuli and cytotoxicity, are very promising mainly for biomedical purposes.

**Funding:** This research received no external funding

**Conflicts of Interest:** The author declares no conflict of interest.

## Abbreviations

| | |
|---|---|
| AA | Acrylic acid |
| AC | Alternating current |
| CA | Casein |
| CS | Chitosan |
| DC | Direct current |
| DP | Degree of polymerization |
| DSC | Differential Scanning Calorimetry |
| G | Graphite |
| Gl | Gelatin |
| GO | Graphene oxide |
| HOMO/LUMO | Energy gap |
| L | Lysine |
| LDPE | Low density polyethylene |
| MW | Molecular weight |
| OPA | Organic phosphonic acid |
| PA | Phytic acid |
| PAA | Poly (acrylic acid) |
| PAANa | Sodium polyacrylate |
| PAM | Polyacrylamide |
| PAN | Polyacrylonitrile |
| PDA-GO | Polydopamine functionalized graphene oxide |
| PDMAEMA | Poly(2-NN′-diethylamino) ethyl methacrylate |
| PE | Polyethylene |
| POFA | Palm oil fuel ash |
| PP | polypropylene |
| PPP | Pomegranate peel powder |
| PU | Polyurethane |
| Pu | Pullulan |
| PVA | Poly (alcohol vinylic) |
| PVAc | Poly (vinyl acetate) |
| PVAM | Malleated PVA |
| PVB | Poly(vinyl butyral) |
| PVP | Poly(vinyl pyrrolidone) |
| RC-SP | Regenerated cellulose softwood pulp |
| SA | Sodium alginate |
| TDM | Total dipole moment |
| TGA | Thermogravimetric analysis |
| TDM | Total dipole moment |

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
