# Peer review of "Poly(Vinyl Alcohol) Recent Contributions to Engineering and Medicine"

_jcs, doi:10.3390/jcs4040175_

Round 1

Reviewer 1 Report

Dear Author

The current communication aims a very important material. The subject and design of the communication are fine but the presentation is extremely poor. Especially introduction and remarks. Also, the author failed to give brief comments on each and every property of PVA which contributes to other matrixes. 

Major comments

The academic English and style of communication need a major improvement.

L-13: Please mention a few of the interesting features.

L-24-25: Repetition, please change the sentence.

L-26: The sentence is not connected with the theme of the first sentence. The author should avoid such a sentence which have not any connection to previous of after sentences. Please put this sentence somewhere suitable.

L-30-32: Please provide suitable references.

L-34: What kind of measures, please mention some.

L-37,38: References, please.

At the end of the introduction please mention why this communication has been written and what it will address in the following sections. 

L-49: Do not use 'They' for hydrogels. Use hydrogels.

L-60,61. Please rephrase this sentence as the current version is not so academic. Please use suitable scientific wording and style to cite other works.

At the end of section 2.1, the author must include his own comments about the current situation of PVA hydrogels and give a short future perspective. All these can be 5-6 lines.

L-73,74: Poorly written please improve.

L-75. Do not start the sentence with the word 'via'.

L-84-86: How the peel powder contributes towards enhancing the film properties. Please elaborate on the related properties here. 

L-91-93: Which property of PVA makes it useful for use in semiconductors. Please give brief reasons for every property which the author mentioned. 

The remarks section is very poorly written. Please improve it.

Reviewer 2 Report

In this study, the author summarized the contributions of poly (vinyl alcohol) to engineering and medicine area. It is an interesting communication manuscipt which may has the guiding significance for biomaterial research in the future.
However, there are a few concerns to be addressed. Please check it carefully.

line 28 "of" before "the size of " should be "on";

line 35 "bio degradation" should be "bio-degradation" or "biological degradation"

line 106 the punctuation of ";" should be ":"

Reviewer 3 Report

Following are some minor changes.

Pp2,80: change ‘dynamo’ to dynamic

Pp2: change dipol to dipole

pp2,80: Thermal parameters indicated that the thermo-chemical ones varied according to PVA/SA composite sites [9]..explain a bit.

Pp3, 121.  selfhealing efficiency was 80% at after 3hours

Pp4, 165, The analysis done pointed… Which analysis

Pp4,185, provide interesting properties.. mention these properties

Consider moving, similar studies together. For examples, all studies where mechanical properties were reported should be together. All studies where absorbent properties of PVA involving dyes can be placed together.

Overall, I find that this draft only reporting facts of cited references. It says the 'what' happened in a particular study but doesn't say 'why' a particular property got improved. what was the cause/science behind it?

It fails to summarize the current state of understanding of PVA. However, it does satisfy the title which is to enlist Recent Contributions of PVA to Engineering and Medicine.

Round 2

Reviewer 1 Report

İt ca be publish now. All my suggestions have been successfully addressed.

Author Response

Many thanks for your suggestions; they really improved my paper.

Reviewer 3 Report

Overall, I suggest answering ‘why’ a certain property improved for every application and acting on the following new comments. Maybe this small review will help readers and get some citations. As a review paper, it should critically analyse best practices of making hydrogels. what research question was being addressed in a given paper/article. 

The current manuscript only reports findings of other researchers without critically examining their quality and their importance.

MW ? add to the table of abr.

Line 42 change The experiment results… The results of the experiment

Line 41 An article studied?... A group of researchers studied

Line49 change, ‘the’ to ‘their’

Line63 ‘biocompatiility’ spelling

Line 65 'Via co-precipitation PVA bio-composites with regenerated cellulose softwood pulp (RC-SP) as a green reinforcement were prepared'. rephrase.

Line 82 ‘a sow release fertilizer was obtained’. What is sow release fertilizer?

Line 88 at a small scale applied, will increase their possibilities to be used in various domains…remove ‘applied’

Line100. Where is the science? Such sentences are fine in introduction.

Line 114 ‘The direct current (DC).conductivity’. remove the fullstop.

Line 120 ‘The dielectric and loss tangent analysis were also carried out [16]’. What was the result

Line 124 ‘The thermal conductivity of AA@G exhibited extremely high enhanced efficiency.’ What does the sentence mean? What is efficiency of thermal conductivity mean?

Does this article talk about thermal efficiency or increase in thermal conductivity?

Line 139 ‘to form stronger H bonds to replace the intermolecular phase of PVA intramolecular interactions [20].’ Article missing in the sentence.

Line 140 what was the use? Why were they developed?

Line 146 what is the end use? What significance is to the reader of this review.

Add a few lines about significance/importance of sponges and membranes and their use.

Line 156 what are the requirements.

Also suggest making subheadings such as membranes, sponges, fuel cells, battery, optics etc.

Line 167/168: sentence needs rephrasing

Line 176 Add TGA/DSC abr. In the list. What is the application of the bio composite for this one and the following in line 180?

Line 194 ‘for more potential applications’ what does this mean?

Line 196 ‘glicerol’ change to glycerol

Line 208-210: meaning of the sentence not clear.

Line 214: ‘are more and more used’ ….choose different words

Define ‘bioactive factor’.

Line 219: Many of them respond to chemical and physical stimuli such as ionic strength, pH, temperature, light, and electrical and magnetic fields. Cite references

Line 221-225: incorrect tense used, was, can be, is all used in the paragraph.

Line 227: mention the application

Line 235: in this section (medicine)until now only 3 research papers were cited. Its is barely an overview. ‘It shows that the’ …..the lines 212 -235 does not show complex demands on organ/tissue development. Please make amendments.

Also, suggest making a table/schematic showing the techniques of PVA hydrogel composites.

Again, remarks section needs expanding.
